# Lipid-Enriched Cooking Modulates Starch Digestibility and Satiety Hormone Responses in Traditional Nixtamalized Maize Tacos

**DOI:** 10.3390/foods14152576

**Published:** 2025-07-23

**Authors:** Julian de la Rosa-Millan

**Affiliations:** Tecnologico de Monterrey, Escuela de Ingenieria y Ciencias, Centro de Biotecnologia FEMSA, Av. Eugenio Garza Sada 2501 Sur, Tecnologico, Monterrey 64849, Nuevo Leon, Mexico; juliandlrm@tec.mx

**Keywords:** tacos, nixtamalized maize, starch digestibility, satiety hormones, lipid–starch interactions

## Abstract

Traditional taco preparation methods, such as oil immersion and steaming, can significantly affect the nutritional and metabolic characteristics of the final product. This study evaluated tacos made with five commercial nixtamalized maize flours and four common fillings (chicharron, beef skirt, potato, and refried beans), processed using three different methods: Plain, Full-Fat, and Patted-Dry. We assessed their chemical composition, starch digestibility, and thermal properties, and measured satiety-related hormone responses in mice. Fillings had a stronger influence on protein, fat, and moisture content than tortilla type. Full-fat tacos exhibited increased amylose–lipid complex formation and a lower gelatinization enthalpy, whereas plain tacos retained more retrograded starch and a crystalline structure. In vitro digestion revealed that Plain tacos, especially those with plant-based fillings, had the highest resistant starch content and the lowest predicted glycemic index. Hierarchical clustering showed that resistant starch, moisture, and gelatinization onset temperature were closely linked in the Plain samples, whereas lipid-driven variables dominated in the Full-Fat tacos. In mice, tacos with a higher resistant starch content led to greater GLP-1 levels, lower ghrelin levels, and reduced insulin responses, suggesting improved satiety and glycemic control. Patted-Dry tacos showed intermediate hormonal effects, supporting their potential as a balanced, health-conscious alternative. These findings demonstrate how traditional preparation techniques can be leveraged to enhance the nutritional profile of culturally relevant foods, such as tacos.

## 1. Introduction

In recent decades, the global burden of obesity, type 2 diabetes mellitus (T2DM), and related cardiometabolic disorders has escalated sharply, driven in part by changes in dietary patterns characterized by high energy density, rapid digestibility, and low fiber intake [1]. A growing body of evidence suggests that the quality of nutritional carbohydrates, rather than quantity alone, plays a significant role in modulating postprandial glycemia, insulin dynamics, and hormonal signals related to satiety and energy intake [2]. In this context, traditional carbohydrate-rich foods must be reevaluated not only for their cultural and sensory value but also for their metabolic impact, considering contemporary health challenges [3].

One key metric for assessing carbohydrate quality is the glycemic index (GI), which ranks foods based on their effect on postprandial blood glucose levels relative to a reference carbohydrate. Low-GI diets have been associated with improved insulin sensitivity, a reduced risk of T2DM, favorable lipid profiles, and enhanced satiety. However, the GI of a food is not fixed; it is highly influenced by factors such as starch structure, processing method, fat content, and interactions within the food matrix [4]. Therefore, understanding how traditional culinary practices alter starch digestibility and hormonal responses is critical for repositioning staple foods within a healthier dietary framework [2].

Among traditional Latin American foods, tacos hold a central place in the daily diet, particularly in Mexico, where variations in tortilla base, fillings, and preparation methods reflect both regional diversity and cultural heritage [5,6]. While tacos are often perceived as street food or comfort food, their nutritional quality varies significantly. The base component, the nixtamalized maize tortilla, is itself the product of a complex alkaline cooking process that modifies starch and enhances mineral bioavailability. Nixtamalization is an alkaline cooking process that alters starch structure and enhances mineral bioavailability in maize. In contrast, oil immersion commonly used in taco preparation can further transform the food matrix by promoting amylose–lipid complex formation and altering the extent of starch gelatinization. These two processes exert distinct effects: nixtamalization modifies inherent starch properties during dough formation, while oil immersion impacts post-cooking structure and digestibility. Understanding these independent contributions is crucial for assessing the impact of traditional methods on glycemic response and hormonal regulation [7,8].

In Mexico, corn tortillas are a central component of the diet, with an estimated per capita consumption of approximately 120–130 g per day, representing a major source of daily carbohydrate intake [2]. However, recent shifts in culinary practices, such as increased use of added fats during taco preparation, have raised concerns about the potential impact on glycemic response and metabolic health. Epidemiological data indicate a rising prevalence of type 2 diabetes mellitus (T2DM) and obesity in populations with high refined carbohydrate intake, particularly in urbanized regions where traditional foods are increasingly modified for taste or convenience [2,3]. These trends highlight the significance of understanding how modifications in tortilla-based foods impact starch digestibility and postprandial metabolism.

Several physicochemical mechanisms influence starch digestibility. Native starch granules undergo gelatinization during cooking, thereby increasing enzyme accessibility and facilitating the release of glucose. However, under certain conditions, particularly in the presence of lipids or during cooling, starch can reassociate into more resistant structures. These include slowly digestible starch (SDS) and resistant starch (RS), both of which are linked to lower postprandial glycemic responses. Notably, resistant starch can take several forms: Resistant starch (RS) refers to starch that escapes digestion in the small intestine and reaches the colon. It is classified into five types: RS1 (physically inaccessible starch, as in whole grains or seeds), RS2 (native granular starch found in raw potatoes or green bananas), RS3 (retrograded starch formed when cooked starches cool and realign), RS4 (chemically modified starches), and RS5 (amylose–lipid complexes formed during cooking in the presence of fats). In traditional taco preparation, particularly when tortillas are immersed in oil and then steamed (“tacos al vapor”), the potential for RS3 and RS5 formation is significant. Cooling promotes retrogradation (RS3), while lipids can interact with amylose to form thermally stable helical complexes (RS5) that resist enzymatic hydrolysis. These structures can lower the rate of starch digestion and reduce postprandial glycemic responses [9,10].

The role of dietary lipids in modulating postprandial metabolism extends beyond their interactions with starch. Lipids delay gastric emptying and stimulate the release of satiety-related hormones such as cholecystokinin (CCK), peptide YY (PYY), and glucagon-like peptide-1 (GLP-1), while attenuating the secretion of ghrelin, a hormone associated with hunger [2,11]. These hormonal effects influence appetite regulation, energy intake, and glucose homeostasis. Therefore, a food that incorporates both structural barriers to starch digestion and favorable hormonal responses may offer synergistic metabolic benefits. However, these mechanisms remain underexplored in traditional food systems such as tacos, where culinary practices have the potential to alter both starch–lipid structure and physiological response [2].

While tacos are often perceived as comfort or street food, their nutritional quality is highly variable. Moreover, tacos are increasingly consumed worldwide, far beyond their traditional roots in Mexico and Latin America, making their nutritional evaluation relevant to a global audience. Although prior studies have explored the sensory and macronutrient aspects of oil-treated traditional foods such as fried rice or chapatis, few have examined how oil immersion specifically alters starch structure and postprandial metabolic responses. In tacos, the immersion–steaming process may uniquely influence both the formation of amylose–lipid complexes and the extent of starch retrogradation features not widely addressed in the previous literature. Our study contributes to this gap by linking traditional preparation techniques to resistant starch profiles and satiety-related hormonal responses. Research on the metabolic effects of tacos has been limited, and studies rarely account for the combined influence of tortilla composition, filling type, and cooking method. Most analyses have focused on macronutrient content or sensory attributes, without examining digestion kinetics or endocrine outcomes [12,13].

In this study, we explore how traditional fat-enriched cooking practices modify the nutritional and physiological behavior of tacos by examining the chemical composition, starch digestibility profile, and postprandial hormonal responses of tacos prepared with five commercial nixtamalized maize flours and four typical fillings, chicharron (Ch), beef skirt (BS), potato mash (P), and refried beans (B). These tacos were prepared using three distinct processing methods: Plain (no oil), Full-Fat (oil immersion and steaming), and Patted-Dry (oil immersion followed by blotting) to assess the influence of surface lipid retention and matrix composition.

We hypothesized that the combination of resistant starch content and lipid interactions would affect both the glycemic potential and the endocrine responses to taco consumption. Specifically, we anticipated that higher RS content in Plain tacos and increased amylose–lipid complexation in Full-Fat tacos would correlate with favorable satiety hormone profiles, including elevated GLP-1, reduced ghrelin, and attenuated insulin responses. We further examined how filler composition modulates these effects, particularly in the case of refried beans (B), which are naturally high in fiber and polyphenols known to lower glycemic response.

By integrating biochemical, structural, and physiological measurements, this study aims to bridge the gap between traditional food preparation methods and modern nutritional science. Our findings provide new insights into how lipid–starch interactions in culturally relevant foods can be leveraged to modulate glycemic response and satiety, offering a scientific foundation for reformulating or designing traditional foods with targeted metabolic health benefits.

## 2. Materials and Methods

### 2.1. Tortilla Production from Nixtamalized Maize Flours

Five commercially available (HEB supermarkets, Monterrey, Mexico) nixtamalized maize flours, Maseca White (MW), Maseca Yellow (MY), Maseca Blue (MB), Maseca Antojitos (MA), and Maseca Tamal (MT), were used as substrates for traditional tortilla making. Flours were selected to represent different pigmentations (white, yellow, blue), particle sizes, and industrial applications (tortillas, tamales, snacks), making them representative of the leading commercial nixtamalized flour types available in Mexico. Each flour was handled according to the manufacturer’s hydration guidelines, with doughs standardized at 55% hydration and rested for 30 min before processing.

Tortillas were produced using a pilot-scale tortilla machine (Villamex Model V25 CB, Tortimaq, Zapopan, Mexico). Dough was sheeted and cut into 16 cm diameter disks (~24 g each), forming tortillas approximately 1.2 mm thick. Cooking was carried out in three stages at 245 °C: 20 s on the first side, 20 s on the second, and a final 20 s puffing stage on the original side. Once cooled to room temperature (25 °C), tortillas were packed and stored at 4 °C until analysis. Moisture, crude protein, fat, ash, and fiber contents were determined using AACC approved methods 44-01.01, 46-13.01, 30-20.01, 08-01.01, and 32-10 [14]. All proximate composition values are expressed on a dry basis.

### 2.2. Taco Fillings and Assembly

Four types of fillings were prepared to reflect common and nutritionally distinct options used in Mexican street tacos. Chicharron, rendered and pressed pork skin and meat, pan-fried until crisp and reheated with guajillo chili sauce. Beef skirt (B), thinly sliced and grilled beef, seasoned lightly with salt. Potato mash: boiled white potatoes (*Solanum tuberosum*, Alpha Variety) mashed with a small amount of vegetable oil and salt. Refried beans, cooked and mashed pinto beans, sautéed in vegetable oil with garlic and onion. These four fillings were selected to reflect commonly consumed taco ingredients in Mexican street and home cooking, offering cultural representativeness and nutritional diversity. Chicharrón and beef skirt represent animal-based, high-protein options with varying fat content. At the same time, refried beans and mashed potatoes provide plant-based alternatives that differ in their fiber and starch content. This variety enabled us to assess how the protein, lipid, and carbohydrate compositions interact with the tortilla matrix and processing method. Fillings were prepared in bulk (2 kg each) and portioned consistently (15 g ± 0.5 g) for each taco. All ingredients were sourced locally from commercial suppliers (HEB Supermarkets, Monterrey, Mexico) and prepared fresh before assembly into tacos. 

Taco samples were prepared by combining standardized portions of tortillas (approximately 40 g per piece) and fillings. Each taco was assembled by placing 15 g of filling (Chicharron, Beef Skirt, Potato, or Beans) in the center of a single tortilla, folding it over to create a half-moon shape, and sealing it gently by hand pressure. The tacos were then subjected to one of the following processing methods:Plain (P) Tacos (used as control), without additional lipid treatments.Full-Fat (FF) Tacos (traditional vapor method): Assembled tacos were immersed briefly (3 s per side) in vegetable soybean cooking oil (190 °C) (Nutrioli, San Pedro Cholula, Mexico) and then steamed in a pot-steamer (double-chamber steamer 30 cm in diameter by 60 cm height) for 15 min to replicate the “tacos al vapor” technique.Patted-Dry (PD) Tacos: Separate batches of Full-Fat tacos were removed from the pot steamer after the process described above, then immediately blotted for 5 s on each side using paper towels (Membersmark, Monterrey, Mexico) to remove excess surface oil. This approach was designed to mimic traditional consumer practices, aiming to avoid excessive lipid consumption.

Each experimental batch was made in triplicate.

### 2.3. Freeze-Drying and Milling of Taco Samples

After each treatment, all samples were immediately submerged in liquid nitrogen for 5 min. Following processing, they were portioned, sealed in airtight polyethylene bags, and stored at −70 °C for 24 h. Samples were then freeze-dried using a laboratory-scale freeze dryer (Labconco FreeZone 6, Kansas City, MO, USA) under the following conditions: shelf temperature, 25 °C; vacuum pressure, ≤0.1 mbar; and drying duration, 48 h. Freeze-drying was performed to ensure the removal of moisture while preserving the thermal and structural integrity of the food matrix. Once thoroughly dried, samples were equilibrated to room temperature in a desiccator for 2 h before milling. Dried taco samples (tortilla and filling combined) were coarsely broken and milled using a blade grinder (IKA A11 Basic Analytical Mill, Werke-Switzerland, Baar, Switzerland) at 18,000 rpm for 60 s in 25 g batches. Between batches, the grinding chamber was cleaned thoroughly with ethanol and allowed to dry to prevent cross-contamination. The resulting powders were passed through a stainless-steel mesh sieve (≤250 µm; ASTM No. 60) to ensure uniform particle size suitable for downstream analyses, including starch digestibility, thermal transitions, phenolic extraction, and hormone response modeling. Milled samples were stored in airtight, opaque containers at room temperature and used within 7 days until analysis to prevent oxidative or moisture-induced changes.

### 2.4. Proximal Composition Analysis

Moisture, protein, lipids, and ash were determined according to AACC methods 44-15.02, 960.52, 32-10.01, and 08-17.01, respectively [14]. Carbohydrates were calculated by difference. The total, insoluble, and soluble dietary fiber contents were measured using the Megazyme K-TDFR Kit (Megazyme, Wicklow, Ireland), adhering to the protocols outlined in AOAC Method 991.43 [15] and AACC Method 32-07.01, respectively. All the experiments were conducted on a dry basis.

### 2.5. Total Phenolic Compounds

Five grams of the sample were dispersed in distilled water and stirred for 5 min at 90 °C. The extract was then filtered using Whatman No. 1 filter paper and centrifuged at 1000× *g* for 15 min. The supernatant was stored in amber-colored bottles at 4 °C until it was used. Total phenolic compounds (TPC) were quantified using the Folin–Ciocalteu method. The extract was mixed with sodium carbonate and Folin–Ciocalteu reagent and then incubated. The absorbance of the complex was measured at 620 nm, and the phenolic content was calculated using a standard curve derived from gallic acid. 

### 2.6. Starch and Amylose Contents

Total starch (K-TSTA) and amylose (K-AMYL) contents were determined using Megazyme kits (Megazyme, Wicklow, Ireland) with 500 mg of sample, following the method reported by de la Rosa-Millan et al. [10]. The total starch procedure is based on the amylolysis of starch granules using a thermostable α-amylase, which renders soluble branched and unbranched maltodextrins. Later, α-glucoamylase converts these molecules into glucose, which is quantified via the glucose oxidase/peroxidase enzyme (GODPOD). In the case of amylose quantification, the starch samples were dissolved and reacted with Concanavalin-A (Con-A), which selectively binds to the reducing end of amylopectin molecules, creating a complex that facilitates precipitation. Later, the supernatant was recovered and hydrolyzed with a mixture of α-amylase and α-glucoamylase, resulting in the quantification of glucose molecules using the GODPOD method.

### 2.7. Thermal Properties and Crystallinity Characteristics

All samples were analyzed using a differential scanning calorimeter (Diamond DSC, PerkinElmer, Norfolk, VA, USA) under the conditions and procedure described by de La Rosa-Millán et al. [10]. The onset temperature (To), peak temperature (Tp), conclusion temperature (Tc), and gelatinization enthalpy (ΔH) were calculated using Pyris software (V 4.02) (PerkinElmer, Norwalk, CT, USA).

The crystalline order was assessed by X-ray Diffraction. The diffraction patterns of starches were obtained using an Advanced D8 diffractometer (Bruker, Coventry, UK) at 35 kV with a CuK-α radiation source (1.542 Å). The samples were scanned in the angular range of 4–35° (2θ). The crystallinity percentage (%C) was determined from the diffractograms by calculating the area corresponding to the crystalline peaks (Ap); from the difference between the area under the curve and the area of the amorphous halo, the total area under the curve (At), and the instrumental noise (N) according to Equation (1):%C = Ap/(At − N)(1)

### 2.8. Molecular Characteristics of Starch and Amylopectin Chain Length Distribution

For this analysis, the samples (500 mg) were dissolved in 90% DMSO. The suspension was mechanically stirred for 1 h in a boiling water bath and then maintained at 25 °C for an additional 24 h. An aliquot (2 mL) of the dispersion was mixed with four volumes of ethanol (8 mL) to precipitate starch. The insoluble starch fraction was separated by centrifugation at 7000 rpm for 20 min, redissolved in 5 mL of hot water (to obtain a concentration of 4 mg/mL), and stirred for 30 min in a boiling water bath. The samples were filtered through a 5.0 µm pore size nylon membrane for chromatographic analysis. In all cases, no visible residues were attached to the syringe filters.

The weight-average molar mass and z-average gyration of amylopectin were determined using high-performance size-exclusion chromatography equipped with multi-angle laser-light scattering and refractive index detectors (HPSEC-MALLS-RI). The HPSEC system consisted of a Waters 1525 binary pump (Waters Corp., Milford, MA, USA), a multi-angle laser-light scattering detector (Dawn, Wyatt Tech. Co., Santa Barbara, CA, USA), and a Waters 2414 refractive index detector. A Shodex OH Pak KB-guard column and an SB-806 HQ column (Showa Denko KK, JM Science, Grand Island, NY, USA) separated amylopectin from amylose. The flow rate used was 0.5 mL/min and the sample concentration was 4 mg/mL. The debranching of the starch was carried out using 15 U of isoamylase (E-ISAMY, Megazyme, Wicklow, Ireland), followed by a 6 h incubation under constant stirring at 45 °C. The enzyme was inactivated by boiling the samples in a water bath for 20 min. Immediately after debranching, the solution was filtered on a 0.45 µm pore size nylon membrane and injected into a CLARET-IR system consisting of a Waters 1525 binary pump (Waters Corp., Milford, MA, USA), a Waters 2414 refractive index detector, and two HR 10/30 columns connected simultaneously, the first containing Superdex 200. The second was performed using Superdex 30 gel (Amersham Biosciences, Piscataway, NJ, USA) at a flow rate of 0.4 mL/min, with deionized water (18 MΩ) as the mobile phase. Pullulan standards with known molecular weights were used (180, 738, 5900, 11,800, 22,800, 47,300, and 112,000 g/mol) (Agilent, Santa Clara, CA, USA). Each sample was analyzed in duplicate using the Empower 3 software (Waters Corp.).

### 2.9. Starch Digestion Fractions and In Vitro Predicted Glycemic Index

In vitro starch digestion fractions were determined according to the method described by Englyst et al. [16]. A 400 mg sample underwent rehydration with 10 mL of deionized water, followed by thermal treatment at boiling temperature for 20 min with intermittent vortex mixing: after cooling to 37 °C, enzymatic hydrolysis commenced by adding 8 mL of a pepsin solution (5.21 mg/mL) and maintaining the mixture in a shaking water bath at 37 °C for 2 h. This was followed by digestion using sodium acetate buffer (0.5 M, pH 5.2) and a combination of enzymatic solutions, including pancreatin, amyloglucosidase, and invertase. Aliquots were taken at 20 and 120 min, and glucose concentration was measured using a glucose oxidase-peroxidase assay. Starch fractions were categorized as rapidly digestible starch (RDS), slowly digestible starch (SDS), and resistant starch (RS) based on their hydrolysis rates using Equations (2)–(4):RDS (%) = (G20 − FG) × 0.9 × 100/TS(2)SDS (%) = (G120 − G20) × 0.9 × 100/TS(3)RS (%) = 100% − SDS − RDS(4)
where

G20 and G120 are glucose concentrations at 20 and 120 min, respectively;FG0 = free glucose;TS: total starch content.

The estimated starch glycemic index (pGI) was validated in vitro using the protocol developed by Goni et al. [4], which simulates the initial phases of starch digestion in the human gastrointestinal tract, as outlined in Equation (5). White bread (wheat-based; Bimbo^®^, Monterrey, Mexico) was used as the reference food. It was freeze-dried, milled to pass a 250 μm opening mesh, and subjected to the same in vitro hydrolysis protocol as taco samples to calculate the hydrolysis index (HI)pGI = 39.71 + 0.549 (HI)(5)
where

HI = Hydrolysis Index, calculated as the area under the hydrolysis curve relative to white bread;pGI = predicted glycemic index.

### 2.10. Evaluation of Postprandial Hormonal Responses in Mice

Postprandial hormonal responses were evaluated in male C57BL/6 J mice (n = 5 per group, 8–10 weeks old), which were selected to minimize variability in hormonal responses and align with previous metabolic studies. Each mouse received a single oral bolus of a taco homogenate with the lowest, medium, and highest starch (RS) content from each procedure (Plain, Full-Fat, and Patted-Dry), regardless of the flour type. Mice were housed under controlled environmental conditions (22 ± 2 °C, 12 h light/dark cycle) with ad libitum access to standard chow and water, except during fasting and testing periods. All procedures were approved by the Institutional Animal Care and Use Committee (RODON Research protocol no. 662, approved in 10 March 2025).

Before testing, mice were fasted for 12 h with free access to water. Each mouse received a single oral bolus of a taco homogenate. The taco homogenates were prepared in distilled water at a 1:1 (*w*/*w*) ratio, resulting in a final moisture content of approximately 50–55%. This hydration level ensured consistency across samples and facilitated enzymatic access during digestion, which may influence the release and timing of satiety-related hormones. Homogenates were standardized to deliver 2 g/kg body weight of available carbohydrate, via gavage. The test formulations represented a range of RS contents (6–14%) and were matched for energy content where possible. Blood samples (~100 µL) were collected via the tail vein at 0 (baseline), 15, 30, 60, and 120 min post administration using EDTA-coated capillary tubes containing DPP-IV inhibitor and aprotinin to stabilize active GLP-1.

Plasma was separated by centrifugation (2000× *g*, 10 min, 4 °C) and stored at −80 °C until analysis. Plasma concentrations of active glucagon-like peptide-1 (GLP-1), acylated ghrelin, and insulin were quantified using validated commercial ELISA kits specific for mouse samples, following the manufacturer’s instructions. All samples were run in duplicate, and intra-assay CVs were maintained below 10%.

Hormonal response profiles were analyzed by calculating the incremental area under the curve (iAUC) over the 2 h using the trapezoidal rule. These iAUC values were used as the primary outcome indicators and reported as mean ± standard deviation for each treatment group. Hormonal trends were interpreted in relation to resistant starch content and predicted glycemic index values determined from in vitro digestion assays.

### 2.11. Statistics and Data Analysis

Differences among flour types, fillings, and taco preparation methods were analyzed using one-way ANOVA with Tukey’s post hoc test (*p* < 0.05 considered significant). Additionally, hierarchical clustering analysis was performed to visualize the data structure and group samples based on their similarities using Euclidean distance and Ward’s analyses. Lastly, a Pearson correlogram was performed on the hormonal response data in relation to selected aspects of digestion and structure.

All analyses were performed using Minitab 21 Statistical Software.

## 3. Results and Discussion

### 3.1. Tortilla Chemical Composition

The proximate analysis of tortillas (Table 1) prepared from five commercial nixtamalized maize flours revealed notable differences in moisture, protein, lipid, and ash content, reflecting both the botanical origin and industrial processing characteristics of each flour. Moisture content ranged from 42.7% to 46.2%, with Maseca Tamal tortillas exhibiting the highest values. This may be attributed to the coarser grind and higher water-binding capacity of tamal-specific flours, which are designed for steamed applications [17]. Higher moisture is relevant not only to texture and shelf life but also to digestibility, as increased hydration facilitates gelatinization and enzymatic access during digestion. Protein content was relatively consistent across flours (~6%), with Maseca Yellow showing slightly higher levels, possibly due to varietal differences in endosperm composition. Though modest in absolute terms, these protein levels may still influence satiety and postprandial hormonal responses, especially in formulations that combine tortillas with high-protein fillings such as chicharrón or beef. Lipid content ranged from 3.7% in Maseca White to 4.5% in Maseca Tamal, with the latter potentially benefiting from the preservation of endogenous lipids during its less refined milling process [18]. Lipids are known to play a dual role in nutrition, contributing to energy density and also interacting with amylose to form complexes (RS5) that can reduce starch digestibility. Ash content, a proxy for mineral content, was highest in Maseca Blue and Maseca Tamal, which may reflect either the use of more mineral-rich maize varieties or retention of pericarp material during nixtamalization. Lastly, the calculated carbohydrates were not statistically different from their starch content, which was approximately 78%. These differences may influence glycemic response, particularly when tortillas are paired with fillings of varying starch quality [19].

### 3.2. Molecular Characteristics of Starch in Tortillas

The molecular architecture of amylopectin across tortillas reveals important distinctions that influence starch functionality, enzymatic susceptibility, and ultimately postprandial metabolic effects. Maseca Yellow and Maseca Tamal exhibited the highest molecular weights (1.76 and 1.72 × 10^8^ g/mol, respectively), indicating greater polymerization and potentially more complex branching structures (Table 2). Interestingly, Maseca Tamal also displayed the highest molecular density (22.64 g/mol/nm^3^), despite having one of the smallest radii of gyration (196.33 nm), suggesting a more tightly packed amylopectin conformation, which may impact gelatinization and retrogradation dynamics. In contrast, Maseca Antojitos had the lowest molecular weight (Mw) (1.22 × 10^8^ g/mol) and density (12.82 g/mol/nm^3^), along with the highest Rz (211.74 nm), indicating a looser and more open molecular structure that may facilitate faster enzymatic access and digestion [20]. The branch chain distribution was relatively consistent across samples, dominated by B1 chains (DP 13-24), followed by B2 and B3 chains. Maseca Blue had the highest proportion of B2 chains (34.37%) and the lowest proportion of B3 chains (7.05%), which is associated with slower retrogradation and greater digestibility due to the formation of fewer long chains that can form double helices. Short chains (A chains, DP 6-12) were remarkably stable across flours (~18%), with minor variations that may influence crystallinity but are less likely to impact digestibility. Overall, the observed differences in molecular architecture suggest that Maseca Tamal’s dense, compact structure might confer higher thermal resistance and slower digestion.

In contrast, with its lower density and more extended conformation, Maseca Antojitos may exhibit greater susceptibility to hydrolysis factors that become crucial when these flours are subjected to traditional fat-rich cooking methods, such as taco steaming or dipping [21]. The molecular weight and structural parameters were consistent across replicates, with coefficients of variation below 10%, indicating good analytical reproducibility. These differences in amylopectin architecture may influence enzyme access, thereby impacting starch digestibility and retrogradation behavior during processing.

### 3.3. Tortilla Thermal Characteristics

The thermal and structural behavior of tortillas made from different maize flours reflects the influence of prior processing and the structural transformations that occur during dough preparation and baking [17]. Compared to native flours, all tortilla samples displayed significantly reduced gelatinization enthalpy (∆H) (Table 3). Maseca Tamal exhibited the lowest value (2.85 J/g), suggesting a more advanced gelatinization state or greater starch damage during cooking. Yet, the onset and peak temperatures shifted to lower values (44.68–49.55 °C), likely due to partial gelatinization and the presence of retrograded starch formed during the cooling phase [19]. Among the samples, Maseca Antojitos exhibited the highest crystallinity (22.35%), possibly due to higher levels of retrograded starch, which is known to resist enzymatic hydrolysis and may contribute to a lower glycemic response in vivo.

On the other hand, Maseca Blue and Maseca Tamal exhibited the lowest crystallinity (13.14% and 13.50%), indicating more amorphous starch structures, which may correlate with faster digestion, depending on the fiber and lipid content [22]. These results highlight how tortilla production induces significant structural reorganization in starch, with varying degrees of retrogradation and residual crystallinity depending on the flour source. These changes are metabolically relevant, as they may modulate starch digestibility and hormonal responses associated with satiety and glycemic control when tortillas are consumed in traditional dishes, such as tacos.

### 3.4. Taco Filling Composition

The nutritional and functional profiles of the taco fillings display distinct macronutrient and bioactive compositions that are likely to influence both metabolic outcomes and consumer satiety [6,8]. Chicharron stands out as the most energy-dense filler, with high levels of protein (32.39%) and lipids (32.38%), and relatively low moisture (25.62%), making it a typical high-fat, high-protein matrix often associated with slower gastric emptying and prolonged satiety (Table 4). In contrast, beef skirt presented a moderate protein level (24.78%) with lower fat (11.92%) and a significantly higher moisture content (60.59%), which may result in faster digestion and reduced energy density per gram. The highest phenolic content was observed in beef skirt (87.92 mg GAE/100 g), which may offer some antioxidant benefits, albeit significantly lower than those of plant-based options. Potato mash, although carbohydrate-rich (15.86%) and moisture-dense (78.55%), is nutritionally poor in protein (2.29%) and lipids (1.61%), suggesting a lower satiety potential and likely a higher glycemic response, particularly when consumed in combination with high-glycemic index (GI) tortillas. Fried beans, however, provide a more balanced macronutrient profile, with moderate protein (9.79%), fiber-containing carbohydrates (12.52%), and low fat (4.28%). Notably, they contained the highest total phenolics (245.17 mg GAE/100 g), consistent with the literature, which shows that legumes, particularly when minimally processed, deliver substantial antioxidant and glycemic-lowering effects [23]. The observed phenolic content, particularly in bean and beef skirt fillings, may contribute to the antioxidant potential of the tacos. However, thermal processing steps such as frying and steaming can reduce phenolic stability through oxidation or degradation, potentially diminishing their bioactivity. Despite this, the presence of residual phenolics may still modulate starch digestibility and glycemic response, either directly or through synergistic effects with fiber and resistant starch. Altogether, the variation in macronutrient and phenolic content among these fillings highlights the nutritional diversity of traditional tacos. Fillers like chicharrón may promote satiety due to their protein and fat content, while bean-based options may modulate glycemia through fiber and polyphenol-mediated mechanisms. Such differences are crucial for understanding the metabolic effects of tacos, particularly when considering the composition and processing methods.

### 3.5. Chemical Composition of Tacos Across Processing Treatments

Figure 1 summarizes the chemical composition of tacos by flour type and filling. To improve readability across the 20 sample combinations and three treatments, we used symbol markers rather than overlapping lines or bars. The full legend is provided for interpretation. Moisture content (■) was consistently the most abundant component across all samples, with notably higher values in tacos filled with plant-based ingredients such as potato and beans. This trend can be attributed to the higher water-binding capacity and hydrophilic nature of these fillers, which retain more moisture during processing.

In contrast, tacos filled with chicharrón exhibited comparatively lower moisture levels, likely due to the high fat content and lower water activity in fried or rendered meat components. Protein content (◆) showed significant variation across samples, with the highest levels observed in tacos containing animal-derived fillings such as beef skirt and chicharrón [24]. These ingredients are naturally rich in muscle proteins and contribute to the overall amino acid density of the product. On the other hand, tacos with potato or bean fillings had substantially lower protein content, reflecting the lower protein concentration of plant-based sources and the absence of animal tissue [25].

Lipid content (▲) varied widely depending on the filling, with chicharrón-filled tacos showing the highest fat levels, which aligns with the fatty nature of this ingredient. Beef skirt tacos had intermediate lipid values, while plant-based fillings, such as potatoes and beans, consistently had low fat content, resulting in a lower energy density [26]. These differences are particularly relevant when considering the impact of dietary fat on satiety and postprandial responses. Ash content (✕), representing the total mineral content, was relatively stable among the different taco types. However, slightly higher values were detected in samples with chicharrón and beans, suggesting the presence of mineral-rich components or condiments used in their preparation [18].

Carbohydrate content (+) demonstrated an inverse trend relative to protein and lipid levels. Tacos filled with potato or beans were richer in carbohydrates, primarily due to their high starch content. This trend was most evident when these fillings were combined with standard tortilla bases, resulting in a carbohydrate-dense matrix. In contrast, tacos with meat-based fillings had lower carbohydrate content due to dilution by protein and fat fractions. These shifts in macronutrient distribution not only affect the nutritional value of the tacos but also their expected behavior during digestion and metabolism [27]. For example, the combination of high lipid and starch levels in chicharrón tacos may promote the formation of amylose–lipid complexes, which can influence digestibility and glucose release. Conversely, the higher moisture and carbohydrate content of bean and potato tacos may affect starch gelatinization and enzyme accessibility, impacting glycemic response.

While the macronutrient and phenolic content of taco fillings varied considerably, this variability was consistent across all flour types and preparation methods. Thus, although the composition of the fillings influenced absolute nutrient values, the design allowed us to isolate the interactive effects of maize flour type and cooking treatment. We acknowledge, however, that this introduces some complexity in interpreting individual variable contributions.

### 3.6. Thermal Characteristics

Figure 2 summarizes the thermal properties of taco samples as determined by differential scanning calorimetry (DSC), with a focus on starch gelatinization and amylose–lipid complex transitions. The peak temperature of gelatinization (Tp, ■) varied moderately across samples, generally falling within the 65–75 °C range. These differences reflect the origin and processing conditions of the starch in the tortilla matrix [12,28]. Notably, tacos incorporating legume-based flours (e.g., lentil or chickpea) exhibited slightly higher Tp values, suggesting a more resistant starch structure or possible interference from protein-starch interactions. In contrast, tortillas made from nixtamalized maize tended to show lower Tp, consistent with the partial gelatinization and starch modification induced by alkaline cooking. The gelatinization enthalpy (ΔH, ▲), indicative of the energy required to disrupt the crystalline regions of starch granules, was lower in samples with high moisture and fat contents, particularly those filled with chicharrón. This reduction in ΔH suggests a partial gelatinization during cooking and possible interference of lipids or proteins with starch swelling and water absorption [26,29].

Regarding amylose–lipid interactions, the Tp of the amylose–lipid complex (Tp AL, ◆) was consistently observed at higher temperatures, typically above 90 °C, confirming the formation of helical inclusion complexes that require greater thermal energy to dissociate [21]. These transitions were more pronounced in tacos containing lipid-rich fillings, especially chicharrón and beef skirt, due to the higher availability of hydrophobic molecules capable of interacting with amylose. The dissociation enthalpy (ΔH AL, ✕) of these complexes followed a similar trend, with higher values found in samples that contained both a source of amylose (such as maize or legume flours) and a significant amount of lipid. This suggests that the thermal stability and extent of complexation are not only dependent on the flour source but also on the presence and type of lipid in the filling [24,30]. Interestingly, samples filled with potato or beans tended to show lower ΔH AL values, likely due to their lower fat content and reduced amylose–lipid interaction. In addition, the observed consistency in Tp AL across most samples indicates that once complexes form, their thermal behavior is more conserved, while enthalpy values reveal the extent of interaction.

The thermal behavior of starch and amylose–lipid complexes in tacos is modulated by both the flour matrix and the filling composition. The gelatinization parameters provide insight into the physical state of starch after processing, while the presence and magnitude of amylose–lipid complexes highlight the potential for modified digestibility [31]. The formation of such complexes is known to reduce starch digestibility and slow glucose release, offering a possible mechanism by which taco composition may influence metabolic responses. Thus, these thermal data not only inform us about processing impacts but also potential nutritional implications, particularly regarding the glycemic index and resistant starch formation [9,10].

To further delineate the influence of preparation methods on amylose–lipid complex formation, a comparative summary of Tp AL and ΔH AL values was examined. Full-Fat tacos consistently showed the highest Tp AL and ΔH AL values, indicating more stable and abundant amylose–lipid complexes. Patted-Dry tacos exhibited intermediate values, while Plain tacos showed the lowest, consistent with minimal lipid availability. These patterns align with the thermal and digestibility data, supporting the idea that fat processing modulates complexation behavior across treatments.

Although amylose–lipid complexes (RS5) have been associated with reduced starch digestibility and a slower glycemic response in vitro and animal models, their direct impact on human metabolism remains to be fully established. Existing studies suggest potential benefits, including moderated glucose absorption and increased satiety; however, human intervention trials are needed to confirm these effects. Therefore, while our findings point to promising structural mechanisms, their translation into human health benefits should be interpreted with caution.

### 3.7. Digestibility

Figure 3 presents the resistant starch (RS) content and predicted glycemic index (pGI) of taco samples as affected by processing treatment (Full-Fat, Patted-Dry, and Plain) and filler type. The RS data (Figure 3A) reveal a clear inverse pattern relative to fat removal: tacos subjected to the Plain condition (i.e., fat not retained after frying) exhibited the highest RS values, particularly in samples filled with potato, beans, or other starchy fillers [23,32]. This observation suggests that the removal of surface fat may favor retrogradation or limit the extent of starch–lipid complexation, thereby increasing the formation or preservation of type 3 retrograded starch (RS). In contrast, Full-Fat and Patted-Dry treatments generally resulted in lower RS values, especially in animal protein-filled tacos (e.g., beef skirt, chicharrón), possibly due to the formation of amylose–lipid complexes that hinder retrogradation but simultaneously slow starch digestibility [9,10]. Interestingly, the RS levels remained relatively stable across flour matrices, indicating that filler and fat interaction play a more decisive role in modulating RS content than the tortilla base alone.

In Figure 3B, the predicted glycemic index (pGI) follows a complementary trend. Tacos processed under the Plain condition, which retained less oil, tended to exhibit lower pGI values, particularly when paired with legume-based or potato fillers. These lower pGI values can be attributed to the higher RS content and possibly reduced fat-mediated facilitation of starch digestion [33]. Conversely, Full-Fat tacos showed a notable increase in pGI, especially for samples containing highly digestible matrices such as white maize or wheat flour, coupled with animal fat-rich fillings. This effect likely arises from the dual role of fat: promoting amylose–lipid complex formation (which may reduce enzymatic access) but also enhancing starch gelatinization during cooking, which can increase glycemic response in specific contexts. Patted-Dry tacos generally displayed intermediate pGI values, suggesting that surface oil contributes modestly but not exclusively to glycemic behavior. Importantly, while chicharrón-filled tacos tended to have higher pGI values in the Full-Fat treatment, they also showed slightly improved values in the Plain condition, supporting the idea that excessive surface lipid may exacerbate digestibility [28,34]. These findings also point to a potentially competitive relationship between the formation of resistant starch types RS3 and RS5. In Plain tacos, the absence of excess lipid allows for greater retrogradation during cooling, favoring RS3 formation. Conversely, in Full-Fat tacos, the presence of surface and matrix-integrated lipids appears to enhance RS5 formation via amylose–lipid complexation, which may limit the availability of linear amylose chains needed for retrogradation and thus reduce RS3 levels. This suggests that lipids not only modulate digestibility through complex formation but may also suppress the crystallization processes that promote the formation of resistant starch (RS3). Further exploration of the thermodynamic and structural competition between these two resistant starch pathways could offer valuable insights for optimizing traditional foods based on targeted metabolic outcomes. It is important to note that although processing conditions were expected to influence surface and internal lipid retention, direct quantification of fat content across treatments was not performed. This limits our ability to correlate lipid retention with RS5 formation or glycemic impact precisely. Future studies should incorporate gravimetric or spectroscopic lipid analysis better to characterize the effect of oil uptake and removal.

Together, RS content was significantly higher in Plain tacos compared to Full-Fat and Patted-Dry treatments (*p* < 0.01), particularly in samples filled with potato or beans. Conversely, predicted glycemic index (pGI) values were lowest in Plain tacos and highest in Full-Fat samples (*p* < 0.05), consistent with their respective RS levels and thermal behaviors. Intermediate values observed in Patted-Dry tacos suggest a partial modulation of digestibility by surface oil removal. These findings reinforce the importance of considering not just the ingredients but also the processing method when designing or recommending traditional foods for health-conscious applications [35]. They also align with the DSC data discussed earlier, where the formation of amylose–lipid complexes and their thermal behavior are closely linked to the digestibility potential.

### 3.8. Multivariate Relationships Among Structural, Thermal, and Nutritional Properties of Tacos Under Varying Fat Treatments

Figure 4 shows a hierarchical cluster analysis (HCA) performed on the complete set of variables measured in Plain taco samples. This analysis offers insights into the relationships among chemical composition, thermal properties, and functional indicators, including the glycemic index and starch digestibility. The dendrogram reveals three main clusters, highlighting meaningful groupings of related variables. On the left side, the first major cluster groups moisture content, resistant starch (RS), and TP, To. This association suggests that higher moisture may support greater starch retrogradation or reduced digestibility, aligning with the observed increase in RS under Plain conditions. It also implies that moisture plays a role in shifting gelatinization behavior, likely through its influence on starch-water interactions [23].

The second central cluster encompasses amylose content, predicted glycemic index (pGI), and digestibility parameters. These variables reflect the digestive performance of tacos and show a strong interdependence: samples with higher amylose content tend to exhibit lower pGI and reduced starch digestibility. This is consistent with the role of amylose in forming more crystalline or complexed structures that resist enzymatic hydrolysis [29,31]. Notably, thermal parameters such as Tp AL and ΔH AL for amylose–lipid complexes are also found within this cluster, reinforcing the idea that digestibility is influenced not just by chemical composition but also by starch–lipid interactions [31].

The third cluster comprises TPC (total phenolic content) and some thermal parameters, such as Tp and ΔH of gelatinization, which are positioned farther from digestibility metrics. This indicates that while gelatinization plays a structural role, it may not linearly determine the postprandial response, particularly in the absence of fat [26,32].

The dendrogram overall emphasizes the complex interrelationships among taco variables and highlights how digestibility is co-regulated by amylose content, lipid interactions, and thermal behavior, rather than composition alone. The clustering also suggests that targeted nutritional modulation (e.g., lowering pGI or increasing RS) may be achieved by manipulating specific flour–filler combinations that alter these clustered traits. This multivariate approach reinforces the conclusions drawn from individual DSC and pGI/RS analyses. Still, it provides a broader systems-level view of how processing and composition co-shape the functional profile of tacos. These insights are particularly valuable for optimizing traditional foods to meet modern nutritional goals, while also bridging the gap between food structure, processing, and health outcomes [12,29].

Figure 5 presents the HCA of all Full-Fat taco samples, providing a multivariate perspective on how fat presence affects the interrelationships among compositional, thermal, and functional properties. Compared to Plain tacos, the dendrogram reveals a notable shift in how variables cluster, underscoring the central role of lipids in modulating food structure and nutritional behavior. One of the most prominent clusters groups moisture and molecular characteristics related to amylopectin fine structure, indicating that in Full-Fat tacos, water retention, visual attributes, and resistant starch levels are more tightly linked [30].

Another central cluster includes thermal transitions related to amylose–lipid complexes (Tp AL, ∆H AL), amylose content, and pGI, suggesting that in Full-Fat samples, starch–lipid interactions become more central in determining glycemic behavior. The clustering of Tp AL and ∆H AL alongside RS and amylose supports the hypothesis that lipid retention promotes the formation of thermally stable amylose–lipid complexes, which may decrease enzymatic digestibility but also increase complexity in pGI prediction. Interestingly, this cluster also connects with TPC, indicating a relationship between bioactive compound retention and starch–lipid interaction dynamics.

The digestibility cluster remains distinct but now shows a looser association with amylose and RS than in Plain tacos, implying that lipid-driven gelatinization or emulsification effects could override or modulate the direct contribution of starch structure to glycemic response. Furthermore, lipid and protein content cluster separately from starch digestibility and thermal variables, highlighting their role as independent modulators of nutritional behavior, possibly through textural or emulsification effects rather than direct molecular interaction with starch [20,36].

The starch gelatinization parameters (Tp, To, and ∆H) form a relatively isolated cluster, suggesting that in the presence of fat, the classic gelatinization behavior becomes decoupled from digestibility. This can be attributed to the competition between water and fat for starch interaction, as well as the formation of amylose–lipid complexes that delay or suppress standard gelatinization endotherms. These findings reinforce the idea that in fried or fat-rich systems, starch–lipid interactions dominate the thermal behavior landscape.

Overall, the HCA of Full-Fat tacos confirms that lipid content profoundly alters the matrix’s functional dynamics. While amylose–lipid complexation emerges as a dominant feature linking thermal behavior and digestibility, other properties such as antioxidant capacity and color also appear to co-vary with fat retention. This integrated behavior supports the notion that Full-Fat tacos are structurally and nutritionally distinct from their Plain counterparts, emphasizing the need to consider lipid–starch–bioactive interactions when optimizing traditional foods for metabolic health [8,27]. Such insights are particularly valuable for designing reformulated or healthier taco versions that preserve sensory appeal while improving glycemic outcomes.

Figure 6 presents an HCA of Patted-Dry taco samples, offering insights into the intermediate behavior of variables when excess oil is removed, but internal lipid interactions remain. Compared to Plain and Full-Fat tacos, this matrix exhibits both shared and unique clustering patterns, suggesting a transitional structural behavior influenced by partial lipid retention [34]. One of the primary clusters includes moisture content and amylopectin molecular characteristics. In contrast, another includes Tp, To (the onset temperature of gelatinization), and RS (%), which mirrors the grouping seen in Plain tacos. This suggests that moisture remains a strong determinant of starch retrogradation and resistant starch formation, even in the presence of some internal fat. However, in contrast to Full-Fat samples, RS here clusters closer to thermal parameters than to digestibility metrics, indicating that retrogradation is more strongly driven by water-starch interactions than by amylose–lipid complexes under these conditions.

A second central cluster is composed of amylose content, Tp AL, and ∆H AL, which link the presence and thermal stability of amylose–lipid complexes to the inherent starch characteristics of the matrix. This grouping implies that although surface oil has been removed, enough lipid remains within the structure to interact with amylose and form complexes detectable by DSC. Their association with thermal variables suggests that these complexes influence starch transition temperatures and may indirectly modulate digestibility [31]. Notably, pGI is located farther from this cluster compared to the Full-Fat condition, which may indicate a reduced or less consistent effect of lipid on glycemic behavior in patted tacos.

The third cluster may reflect the influence of thermal stability on the SDS fraction, suggesting a compensatory relationship where the molecular characteristics, particularly those of amylose, allow lipid retention and contribute to slower digestibility [21].

Additionally, lipid and protein contents are positioned more independently in this dendrogram, unlike in Full-Fat tacos, where they clustered more tightly with digestibility traits. This detachment suggests that in Patted-Dry samples, the role of macronutrients becomes more compositionally distinct, no longer dominating postprandial indicators. Overall, this dendrogram indicates that Patted-Dry tacos exhibit an intermediate metabolic and structural profile, more complex than Plain samples but less lipid-driven than Full-Fat ones. The variable associations in Patted-Dry tacos highlight a balance between residual lipid–amylose interactions and the preservation of starch–water and polyphenol dynamics. This processing condition may offer a favorable compromise, allowing for sufficient flavor and texture retention while limiting glycemic impact and preserving nutritional quality. These findings support the potential of fat-removal strategies in traditional food formulations to optimize health outcomes without compromising core sensory characteristics [27].

### 3.9. Satiety-Related Hormonal Responses

The hormonal response data presented in Figure 7 illustrate the postprandial behavior of GLP-1, ghrelin, and insulin in mice following administration of taco formulations with varying resistant starch (RS) content and fat treatments. A clear pattern emerges: tacos with higher RS content were associated with elevated GLP-1 levels, reduced ghrelin secretion, and attenuated insulin responses, supporting the metabolic benefits of resistant starch in modulating satiety and glycemic control [2].

Notably, GLP-1 levels were significantly higher in samples such as MA DF Bean and MA DF Beef Skirt, which also had among the highest RS values in the dataset. GLP-1 is an incretin hormone that promotes insulin secretion and enhances satiety by slowing gastric emptying and reducing appetite. The enhanced GLP-1 response in these samples likely results from the slower digestion and fermentation of RS in the distal gut, leading to increased stimulation of L-cells [25].

In contrast, ghrelin, a hunger-promoting hormone secreted mainly in the stomach, was markedly lower in samples with higher RS. This inverse relationship supports the concept that resistant starch suppresses appetite by promoting a longer-lasting sense of fullness. The highest ghrelin levels were observed in low-RS tacos, including MY P Chicharrón and MY P Potato, consistent with their predicted high glycemic index and rapid digestibility.

The insulin response also followed a favorable trend: tacos with greater RS content elicited significantly lower insulin AUCs compared to their low-RS counterparts. This reduction suggests improved glycemic control and a reduced need for insulin secretion, potentially mitigating postprandial insulin spikes and promoting metabolic flexibility [37]. Interestingly, the intermediate responses observed in Patted-Dry tacos (e.g., MY PD Potato) indicate that modest reductions in lipid content may preserve some of the metabolic advantages of RS while maintaining product palatability.

The hormonal response patterns demonstrate that taco formulations enriched in resistant starch, especially when combined with moderate fat reduction, may enhance satiety and reduce postprandial insulin demand. These findings underscore the significance of ingredient selection and processing methods in influencing not only digestibility and glycemic impact but also endocrine responses linked to appetite regulation and metabolic health.

While these findings in mice provide valuable insight into the satiety and glycemic effects of resistant starch and fat interactions, caution is warranted when extrapolating to human physiology. Differences in gut microbiota, hormone receptor sensitivity, and metabolic regulation between species may alter the magnitude or direction of postprandial responses. Nonetheless, prior studies have reported parallel trends in GLP-1 and ghrelin modulation in both rodents and humans following intake of resistant starch [25,37], supporting the relevance of our model. Further human-based trials are necessary to confirm these endocrine responses in real-world dietary contexts.

### 3.10. Correlation Analysis of Nutritional, Structural, and Hormonal Parameters

The correlation matrix provided in Figure 8 illustrates the multivariate relationships among chemical composition, starch structure, thermal properties, and postprandial hormonal responses observed across the taco formulations. As expected, resistant starch (RS) content displayed strong negative correlations with predicted glycemic index (pGI) and insulin levels, confirming its role in attenuating postprandial glycemic excursions. RS was also inversely associated with rapidly digestible starch (RDS) and positively related to total phenolic content (TPC), suggesting that fiber- and polyphenol-rich fillings, particularly those derived from legumes, enhance the functional properties of starch.

Amylose content correlated positively with the enthalpy of amylose–lipid complexes (ΔH AL), highlighting that lipid interactions are enhanced in samples with higher linear starch fractions. Notably, GLP-1 secretion was positively correlated with RS and TPC, whereas ghrelin showed inverse correlations with the same variables, reinforcing the physiological relevance of these structural components for satiety regulation.

Thermal transition variables (Tp AL, ΔH AL) also correlated with slower digestibility markers, supporting the notion that amylose–lipid complex stability contributes to functional starch resistance. Overall, this multivariate analysis confirms that compositional and structural features interact to shape metabolic outcomes, providing a systems-level framework that links food processing to physiological impact.

## 4. Conclusions

The metabolic impact of tacos is significantly influenced by both the choice of filling and the type of fat used during preparation. Tacos with higher resistant starch (RS) content, particularly those using bean or potato fillings in Plain or Patted-Dry formats, were associated with lower predicted glycemic index (pGI), elevated GLP-1, reduced ghrelin, and attenuated insulin responses, reflecting enhanced satiety and improved glycemic control. Thermal and molecular analyses revealed that lipid-rich formulations favored the formation of amylose lipid complexes, which contribute to reduced starch digestibility but may also raise pGI depending on the extent of gelatinization. Hierarchical clustering demonstrated that RS, moisture, and gelatinization onset temperatures co-varied, particularly in Plain tacos. In contrast, in Full-Fat samples, lipid-related variables dominated the functional profile. Importantly, Patted-Dry tacos showed intermediate behaviors, suggesting that partial removal of surface oil may preserve favorable metabolic features without compromising texture or flavor.

Although our findings suggest that lower-fat or moderately fat-reduced taco preparations may support improved glycemic control and satiety, practical implementation may be limited by cultural expectations around flavor, texture, and traditional preparation techniques. Future reformulation strategies should aim to strike a balance between sensory acceptance and nutritional benefits, thereby enhancing the product’s real-world feasibility.

## Figures and Tables

**Figure 1 foods-14-02576-f001:**
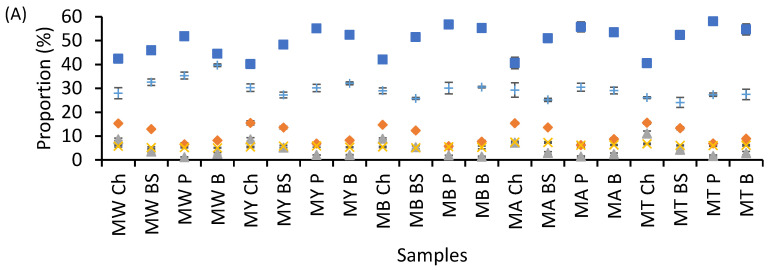
Chemical composition of taco samples across three processing methods: (**A**) Plain, (**B**) Full-Fat, (**C**) Patted-Dry. Each data point represents the mean of triplicate samples. Symbols represent individual chemical components: ■ Moisture, ◆ Protein, ▲ Lipids, ✕ Ash, + Carbohydrates (Calculated by difference). MW: Maseca White, MY: Maseca Yellow, MB: Maseca Blue, MA: Maseca Antojitos, MT: Maseca Tamal. Fillings: Ch—Chicharrón, BS—Beef Skirt, P—Potato, B—Bean.

**Figure 2 foods-14-02576-f002:**
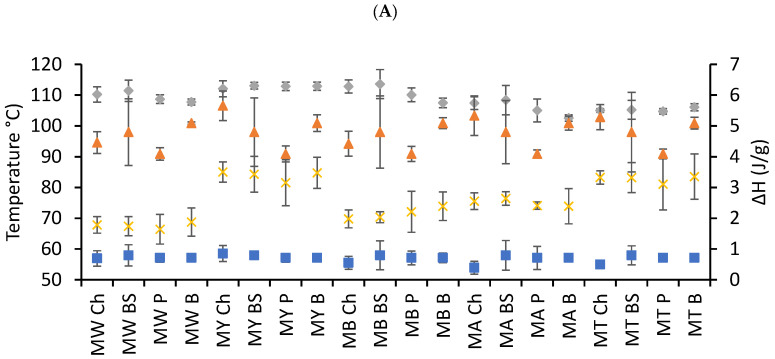
Selected thermal characteristics of tacos. (**A**) Plain, (**B**) Full-Fat, (**C**) Patted-Dry. Tp (■), Tp AL (◆), ΔH (▲), ΔH AL (✕). MW: Maseca White, MY: Maseca Yellow, MB: Maseca Blue, MA: Maseca Antojitos, MT: Maseca Tamal. Ch: Chicharron, BS: Beef Skirt, P: Potato, B: Bean.

**Figure 3 foods-14-02576-f003:**
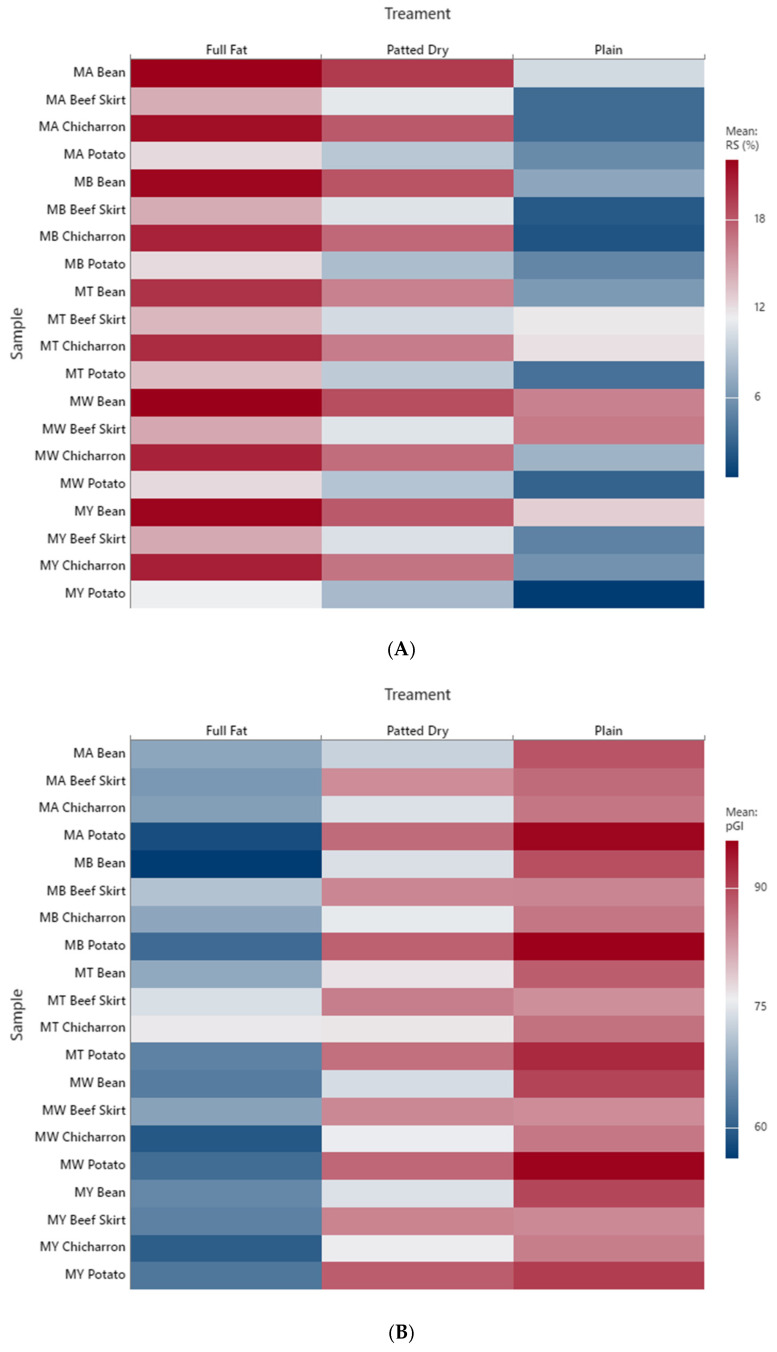
Effect of Lipid Treatment and Filler Composition on (**A**) RS Content and (**B**) Predicted Glycemic Response in Tacos across treatments. MW: Maseca White, MY: Maseca Yellow, MB: Maseca Blue, MA: Maseca Antojitos, MT: Maseca Tamal.

**Figure 4 foods-14-02576-f004:**
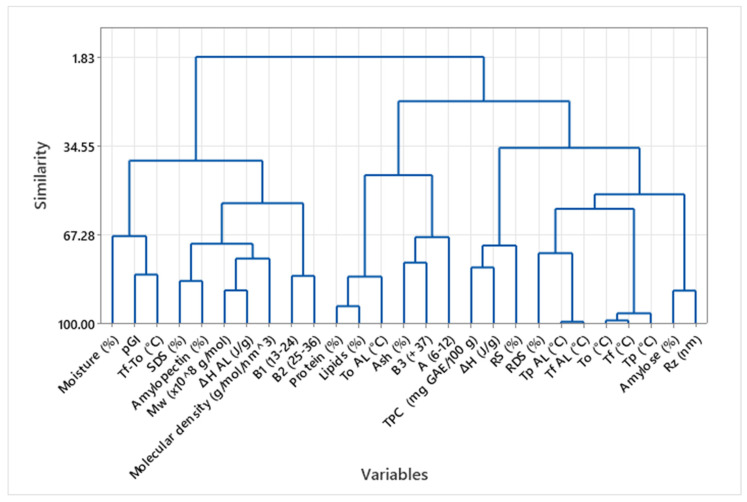
Cluster Analysis of Compositional and Functional Variables in Plain Taco Samples.

**Figure 5 foods-14-02576-f005:**
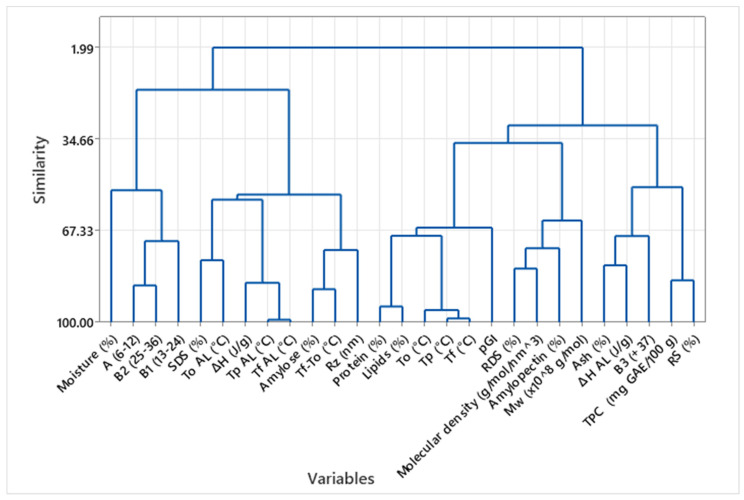
Cluster Analysis of Compositional and Functional Variables in Full-Fat Taco Samples.

**Figure 6 foods-14-02576-f006:**
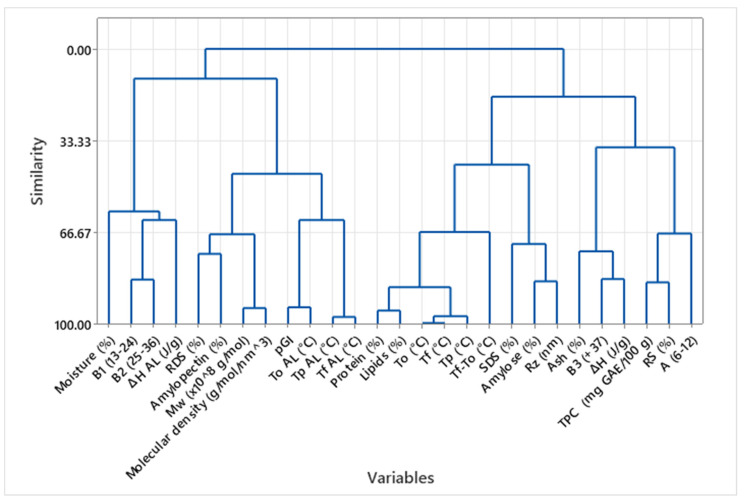
Cluster Analysis of Compositional and Functional Variables in Patted-Dry Samples.

**Figure 7 foods-14-02576-f007:**
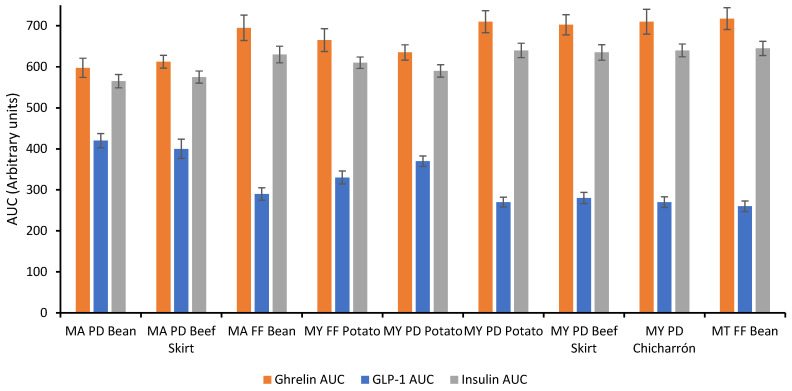
Postprandial Hormonal Responses to Taco Formulations with Varying Fat Treatments and Fillings. MY: Maseca Yellow, MA: Maseca Antojitos, MT: Maseca Tamal. P: Plain taco, FF: Full Fat, PD: Patted-Dry.

**Figure 8 foods-14-02576-f008:**
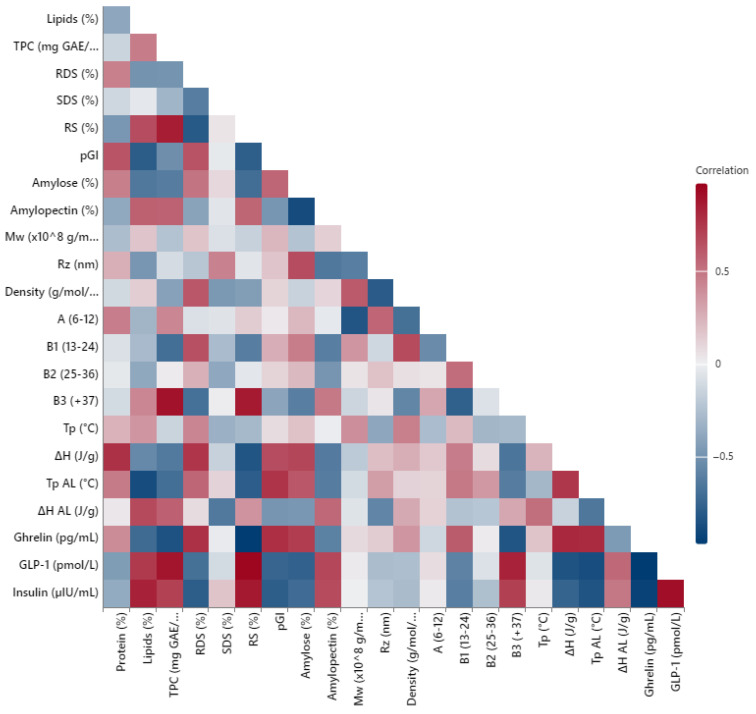
Pearson correlation matrix showing pairwise relationships among chemical composition, starch structure, thermal transitions, and postprandial hormone responses across taco samples. Legend note: Red indicates positive correlations, while blue indicates negative correlations; the stronger the color intensity, the higher the correlation magnitude. Abbreviations: RDS—rapidly digestible starch, SDS—slowly digestible starch, RS—resistant starch, pGI: Predicted glycemic index, TPC—total phenolic content, Mw—molecular weight, Rz—hydrodynamic radius, Tp—gelatinization peak temperature, Tp AL—amylose–lipid complex transition temperature, ΔH—gelatinization enthalpy, ΔH AL—amylose–lipid complex enthalpy.

**Table 1 foods-14-02576-t001:** Proximate Composition of Tortillas Prepared from Five Commercial Nixtamalized Maize Flours.

Tortilla	Moisture (%)	Protein (%)	Lipids (%)	Ash (%)	Carbohydrates (%) *
MW	44.2 ± 1.5 ^a^	6.1 ± 0.4 ^a^	3.7 ± 0.3 ^a^	3.8 ± 0.2 ^a^	86.4 ± 1.2 ^a^
MY	42.7 ± 1.2 ^b^	6.3 ± 0.5 ^a^	3.9 ± 0.2 ^a^	3.0 ± 0.3 ^a^	86.8 ± 0.8 ^a^
MB	45.6 ± 1.7 ^a^	5.8 ± 0.3 ^a^	4.1 ± 0.4 ^a^	3.4 ± 0.2 ^a^	86.7 ± 1.1 ^a^
MA	43.9 ± 1.0 ^b^	6.2 ± 0.6 ^a^	4.0 ± 0.3 ^a^	3.1 ± 0.3 ^a^	86.7 ± 1.5 ^a^
MT	46.2 ± 1.3 ^a^	6.0 ± 0.7 ^a^	4.5 ± 0.5 ^a^	3.3 ± 0.2 ^a^	86.2 ± 1.6 ^a^

Values are mean ± SD. Different letters in the same column indicate significant differences (*p* < 0.05) according to Tukey’s test. MW: Maseca White, MY: Maseca Yellow, MB: Maseca Blue, MA: Maseca Antojitos, MT: Maseca Tamal.* Calculated by difference.

**Table 2 foods-14-02576-t002:** Molecular Weight, Radius of Gyration, Density, and Branch Chain Distribution of Amylopectin from Commercial Tortillas.

Sample	Mw (×10^8^ g/mol)	Rz (nm)	ρ (g/mol/nm^3^)	A	B1	B2	B3
MW	1.51 ± 0.23 ^b^	207.56 ± 2.15 ^b^	17.00 ± 2.99 ^b^	18.38 ± 0.09 ^a^	39.05 ± 0.27 ^a^	32.95 ± 0.50 ^b^	9.62 ± 0.42 ^a^
MY	1.76 ± 0.20 ^a^	207.54 ± 1.84 ^b^	19.72 ± 2.52 ^b^	18.30 ± 0.29 ^a^	40.36 ± 1.09 ^a^	33.19 ± 0.09 ^a^	8.15 ± 0.32 ^b^
MB	1.46 ± 0.07 ^b^	198.85 ± 0.63 ^c^	18.63 ± 1.01 ^b^	18.30 ± 0.43 ^a^	40.28 ± 0.42 ^a^	34.37 ± 0.24 ^a^	7.05 ± 0.40 ^c^
MA	1.22 ± 0.12 ^b^	211.74 ± 1.49 ^a^	12.82 ± 1.41 ^c^	18.40 ± 0.20 ^a^	39.26 ± 0.60 ^a^	32.96 ± 0.36 ^b^	9.38 ± 0.21 ^a^
MT	1.72 ± 0.34 ^a^	196.33 ± 2.73 ^c^	22.64 ± 3.66 ^a^	18.29 ± 0.09 ^a^	39.51 ± 0.71 ^a^	33.01 ± 0.09 ^a^	9.19 ± 0.70 ^a^

Values are mean ± SD. Different letters in the same column indicate significant differences (*p* < 0.05) according to Tukey’s test. MW: Maseca White, MY: Maseca Yellow, MB: Maseca Blue, MA: Maseca Antojitos, MT: Maseca Tamal. Mw: Molecular weight, Rz: Hydrodynamic radius, ρ: Molecular density. Amylopectin chain length: A (DP 6–12), B1 (DP 13–24), B2 (DP 25–36), B3 (DP ≥ 37).

**Table 3 foods-14-02576-t003:** Thermal Properties and Crystallinity of Maize Tortillas Prepared from Commercial Flours.

Sample	To (°C)	Tp (°C)	Tf (°C)	Tf–To (°C)	∆H (J/g)	Crystallinity (%)
MW	49.55 ± 1.30 ^a^	56.95 ± 1.10 ^a^	60.76 ± 1.04 ^a^	11.21 ± 0.27 ^a^	4.46 ± 0.36 ^a^	15.71 ± 1.61 ^a^
MY	47.96 ± 0.47 ^b^	55.51 ± 0.53 ^b^	59.35 ± 0.83 ^a^	11.39 ± 0.67 ^a^	4.58 ± 0.14 ^a^	16.33 ± 1.23 ^a^
MB	46.46 ± 0.46 ^b^	53.79 ± 0.97 ^b^	57.58 ± 1.09 ^b^	11.12 ± 1.39 ^a^	3.77 ± 0.07 ^b^	13.14 ± 1.44 ^b^
MA	46.84 ± 1.96 ^b^	54.60 ± 1.60 ^b^	58.66 ± 1.95 ^a^	11.82 ± 0.99 ^a^	4.26 ± 0.09 ^a^	12.35 ± 1.80 ^b^
MT	44.68 ± 1.13 ^c^	52.03 ± 0.68 ^c^	56.05 ± 1.85 ^b^	11.37 ± 0.78 ^a^	2.85 ± 0.64 ^c^	13.50 ± 1.64 ^b^

Values are mean ± SD. Different letters in the same column indicate significant differences (*p* < 0.05) according to Tukey’s test. MW: Maseca White, MY: Maseca Yellow, MB: Maseca Blue, MA: Maseca Antojitos, MT: Maseca Tamal. To: Onset gelatinization temperature, Tp: Peak gelatinization temperature, Tf: Final gelatinization temperature, ∆H: Enthalpy of Gelatinization.

**Table 4 foods-14-02576-t004:** Nutritional Composition and Phenolic Content of Taco Fillings.

Filler	Moisture (%)	Protein (%)	Lipids (%)	Ash (%)	Carbohydrates (%) *	Phenolics (mg GAE/100 g)
Chicharron	25.62 ± 0.23 ^d^	42.39 ± 1.23 ^b^	32.38 ± 0.53 ^a^	4.79 ± 0.05 ^a^	20.44 ± 0.69 ^d^	67.01 ± 0.50 ^c^
Beef skirt	60.59 ± 0.47 ^c^	44.78 ± 0.12 ^a^	11.92 ± 0.19 ^b^	2.31 ± 0.22 ^c^	40.99 ± 0.40 ^c^	87.92 ± 1.83 ^b^
Potato mash	78.55 ± 0.35 ^a^	2.29 ± 0.10 ^d^	1.61 ± 0.31 ^d^	1.69 ± 0.15 ^d^	94.41 ± 0.24 ^a^	11.85 ± 0.56 ^d^
Bean fried	70.37 ± 0.41 ^b^	9.79 ± 0.07 ^c^	4.28 ± 0.26 ^b^	3.03 ± 0.28 ^b^	82.9 ± 0.48 ^b^	245.17 ± 1.98 ^a^

Values are mean ± SD. Different letters in the same column indicate significant differences (*p* < 0.05) according to Tukey’s test. GAE: Gallic acid equivalents.* Calculated by difference.

## Data Availability

The original contributions presented in this study are included in the article. Further inquiries can be directed to the corresponding author.

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
