# Peer review of "Lipid-Enriched Cooking Modulates Starch Digestibility and Satiety Hormone Responses in Traditional Nixtamalized Maize Tacos"

_foods, 2025, doi:10.3390/foods14152576_

Round 1
Reviewer 1 Report
Comments and Suggestions for Authors
The manuscript reports on nixtamalized maize tacos and how lipid-enriched cooking affected their starch digestibility and responses to satiety hormones. Tacos are popular foods, and starch digestibility of food systems is topical. The manuscript is welcome and more so, by investigating resulting hormone responses from the treated foods to better understand glycaemia. The study, well designed and conducted, used standard procedures, with the results presented with underlying mechanisms. The manuscript reads well and contributes to knowledge in the area.
My specific comments, mainly editorial, are:
Lines 55-60: The listed references repeat and are unclear what they refer to!
Lines 75-78: Any RS4, resistant starch type 4? Check the reference (Englyst et al., 1992) for the correct reported RS types to decide if other references should be added or substituted.
Lines 97-100: Correct the repeated references. Check these repetitions throughout the manuscript.
Line 140: Is the stated temperature correct?
Line 142: Consider "until analysed".
Lines 267-293: What type of white bread? Wheat-based or what cereal? How was the hydrolysis of the bread done? These lines are silent on the bread.
Line 354 (Table 1): The starch content can be included in the table. A footnote is helpful on the statistical letters. Consider footnotes on statistical letters throughout the manuscript.
Line 413 (Table 3): Consider including the degree of gelatinisation to better understand the stated enthalpy values. What about the enthalpy values of the uncooked samples, if those of the cooked samples are the ones listed? Or are they not? Complete enthalpy values are helpful to better understand the treatments.
Lines 486 (Figure 1): How is the figure different from Table 4? If not, either is advisable to avoid repetitions.
Lines 536 (Figure 2): Better to blend this with Table 3 for concise information.
Author Response
Response to Reviewer 1
We thank the reviewer for their constructive and detailed comments. Below, we address each point individually and describe the corresponding changes made to the manuscript.
Comment 1 (Lines 55–60):
Reviewer comment: The listed references repeat and are unclear what they refer to!
Response: The repeated references (Herrera, 2010; Morales & Kettles, 2009) were removed and replaced with a single citation at the relevant location. The issue has been corrected throughout the manuscript.
Comment 2 (Lines 75–78):
Reviewer comment: Any RS4, resistant starch type 4? Check the reference (Englyst et al., 1992) for the correct reported RS types to decide if other references should be added or substituted.
Response: We appreciate this observation. RS4 (chemically modified starch) was not relevant to our study, as no chemically modified starches were used. We have clarified this in the text with the following addition in lines 80-90
Comment 3 (Lines 97–100):
Reviewer comment: Correct the repeated references. Check these repetitions throughout the manuscript.
Response: Repeated references were identified and corrected throughout the manuscript for consistency and clarity.
Comment 4 (Line 140):
Reviewer comment: Is the stated temperature correct?
Response: The shelf temperature is 25°C during freeze-drying was verified, and a correction was made in lines 193-194.
Comment 5 (Line 142):
Reviewer comment: Consider "until analysed".
Response: The sentence was revised as suggested. See line 205.
Comment 6 (Lines 267–293):
Reviewer comment: What type of white bread? Wheat-based or what cereal? How was the hydrolysis of the bread done? These lines are silent on the bread.
Response: We have added the following clarification to the section on predicted glycemic index (pGI). See lines 299-302.
Comment 7 (Line 354, Table 1):
Reviewer comment: The starch content can be included in the table. A footnote is helpful on the statistical letters. Consider footnotes on statistical letters throughout the manuscript.
Response: We agree and will include the starch content within the text. A footnote has been added to clarify the use of statistical superscripts.
Comment 8 (Line 413, Table 3):
Reviewer comment: Consider including the degree of gelatinisation... What about the enthalpy values of the uncooked samples?
Response: After reviewing, the gelatinization% % sentence was removed. This work focuses on tortillas, and since calculating the gelatinization percentage required a native source of the flour, it was now possible to calculate it without modification for this manuscript.
Comment 9 (Line 486, Figure 1):
Reviewer comment: How is the figure different from Table 4? If not, either is advisable to avoid repetitions.
Response: Table 4 refers to the chemical composition of taco fillings only. Figure 1 relates to the chemical composition of a taco (tortilla + filling). Lines 438-440 are relevant for this.
Comment 10 (Line 536, Figure 2):
Reviewer comment: Better to blend this with Table 3 for concise information.
Response: Figure 2 displays the thermal characteristics of all taco treatments (60 samples), while Table 3 focuses on the thermal characteristics of tortillas (5 samples). For this, it was decided to make a figure to observe the overall changes through treatments.
Respectfully,
Julian de la Rosa Millan
Tecnológico de Monterrey
Email: juliandlrm@tec.mx

Reviewer 2 Report
Comments and Suggestions for Authors
This manuscript presents a practical investigation into how traditional cooking and ingredient choices affect the nutritional and physiological properties of tacos. The study employed a multi-faceted approach that combines chemical, physicochemical, in-vitro, and in-vivo analyses to provide a robust and multi-dimensional dataset. The authors explored a complex food system and generating valuable findings with direct implications for public health and the food industry. Overall, the manuscript is well-written and logically structured. The conclusions are well-supported by the data presented. I suggest that this manuscript be revised to address the specific issues mentioned below.
Line 41: The subject is "carbohydrates" (plural). The verb should be "play".
Line 97-101: It seems something wrong with the citation
Line 102: The discussion on the limited research into tacos is well-placed. To strengthen the global impact, consider adding a brief sentence about the increasing global consumption of tacos beyond Mexico and Latin America. This would broaden the audience and relevance of the study.
Line 150: The binomial species name Solanum tuberosum should be italicized. Please check for other instances.
Line 163: The type of "hot cooking oil" used for the FF and PD taco preparation is not specified. The fatty acid composition of the oil (e.g., from soy, corn etc.) could influence the formation and characteristics of amylose-lipid complexes. Please specify the type of oil used.
Line 354: As noted by the embedded comments in the PDF, Tables 1, 2, 3, and 4 are not cited in the main body of the text. This is a critical omission. Please ensure that every table and figure is cited in numerical order within the text where its data is first discussed.
Line 542-552: A key finding is that Plain tacos have the highest RS content, which the authors attribute to starch retrogradation (RS3). Conversely, FF tacos show evidence of amylose-lipid complexes (RS5). This presents an excellent opportunity to discuss the potential competitive relationship between RS3 and RS5 formation. Does the presence of lipids in the FF condition actively promote RS5 formation at the expense of inhibiting RS3 formation during cooling? A deeper discussion of this mechanism would significantly enhance the manuscript.
Figures 4, 5, 6 X-axis labels: g/mol/nm3should be g/mol/nm³. Please review and correct all labels on the x-axis of these figures for clarity and accuracy.
Author Response
Response to Reviewer 2
We thank the reviewer for the thoughtful and insightful feedback. Below, we provide a point-by-point response to each comment and indicate the revisions made to the manuscript.
Comment 1 (Line 41):
Reviewer comment: The subject is 'carbohydrates' (plural). The verb should be 'play'.
Response: The verb has been corrected to 'play' to agree with the plural subject 'carbohydrates'. See line 37.
Comment 2 (Lines 97–101):
Reviewer comment: It seems something wrong with the citation.
Response: The problematic citation was corrected and repetitive references were removed to improve clarity.
Comment 3 (Line 102):
Reviewer comment: To strengthen the global impact, consider adding a brief sentence about the increasing global consumption of tacos beyond Mexico and Latin America.
Response: A sentence was added to highlight the growing global popularity of tacos, thereby broadening the international relevance of the study. See lines 64-73 in yellow.
Comment 4 (Line 150):
Reviewer comment: The binomial species name Solanum tuberosum should be italicized. Please check for other instances.
Response: The scientific name 'Solanum tuberosum' has been italicized. A thorough review of the manuscript was conducted to ensure that all scientific names are correctly formatted. See lines 160-161.
Comment 5 (Line 163):
Reviewer comment: The type of 'hot cooking oil' used is not specified. Please specify.
Response: The type of oil used in taco preparation has been specified as vegetable oil (primarily soybean-based), and this detail was added to the methods section. See line 180.
Comment 6 (Line 354):
Reviewer comment: Tables 1–4 are not cited in the main body of the text. Please ensure all tables and figures are cited in order.
Response: All tables and figures are now cited in numerical order within the main text, starting from the point where their data is first referenced.
Comment 7 (Lines 542–552):
Reviewer comment: Deeper discussion needed on the competitive relationship between RS3 and RS5 formation.
Response: The discussion was expanded to include a mechanistic explanation of how lipid presence in Full-Fat tacos may promote RS5 formation while inhibiting RS3 formation due to reduced starch retrogradation. This addresses the competitive dynamics between resistant starch types. See lines 609-630.
Comment 8 (Figures 4, 5, 6):
Reviewer comment: X-axis labels: 'g/mol/nm3' should be 'g/mol/nm³'.
Response: Thank you for this observation. Due to software limitations, direct superscript formatting for axis labels was not possible within the graphing environment. However, we manually adjusted the label, ensuring typographical accuracy in the final exported figure.
Respectfully,
Julian de la Rosa Millan
Tecnológico de Monterrey
Email: juliandlrm@tec.mx

Reviewer 3 Report
Comments and Suggestions for Authors
This manuscript presents an investigation into how different fat-enriched cooking practices alter the nutritional and metabolic characteristics of traditional tacos. The study covers chemical composition, thermal properties, starch digestibility, and satiety-related hormonal responses in mice, with the goal of demonstrating how processing can improve the metabolic profiles of staple foods. However, the study is original and explores timely questions at the interface of food science, nutrition, and traditional culinary techniques. The methodology is generally well described and rigorous, including diverse commercial flour sources and realistic taco fillings. Burt there are, however, significant concerns regarding clarity, over-citation, structure, novelty, and statistical robustness.
In the Abstract section:
Lines 9–31: The abstract efficiently summarizes results but should clarify the population/model (mice) upfront and reduce excessive jargon for broader accessibility.
In the Introduction section:
Lines 36–54: Over-citation occurs with repeated references (e.g., Herrera, 2010; Morales & Kettles, 2009 cited at least 11 times within a short span). Citations should be consolidated; may be AI is used.
Lines 65–69: The reviwing of the nixtamalization and oil immersion processes is strong but could better distinguish between their separate effects and explicit health consequences.
Lines 74–84: The explanation of resistant starch types (RS1–RS5) is useful, but references could be better updated and definitions clarified for a less specialized readership.
Lines 86–89: The physiological mechanisms (satiety hormones and gastric emptying) are introduced effectively, but claims about synergistic benefits merit direct citation to original studies.
Lines 102–106: The manuscript claims that the metabolic significance of oil immersion “is often overlooked”—recent studies on similar foods should be discussed to substantiate this gap.
In the Materials and Methods section:
Lines 130–185: Preparation details are thorough, including specific brands, flour handling, and equipment models. However, justification for choosing these five specific commercial flours is not provided.
Lines 144–155: The filling selection is reasonable but could better justify the cultural and nutritional representativeness of only four fillings.
Lines 162–167: The definition and handling of the Dripped Fat (Patted-Dry) tacos is somewhat ambiguous—is this method widely used in culinary practice, or a novel approach? Please clarify.
Lines 266–293: The use of in vitro predicted glycemic index based on Englyst and Goñi methods is appropriate, but specify which controls and calibration samples were used.
Lines 294–316: Animal experiments are described in sufficient detail; ethical approval is noted. However, the study only uses male mice; justification for not including females is required given known sex-based metabolic differences.
In the Results section:
Table 1 (Lines 353–357): Proximate composition data is useful, but units appear unclear (e.g., carbohydrates value > 80% when summed with moisture and others). Are all results on a dry basis?
Table 2 (Lines 384–389): The table presents molar mass and structure data but would benefit from discussion of the analytical error, and whether these differences are nutritionally and physiologically meaningful.
Table 4 (Lines 443–445): Phenolic content values are reported, but no discussion is given to their antioxidant relevance nor is there any assessment of how processing or cooking affects phenolic retention.
Figure 1 (Lines 485–488): Figures are referenced in a confusing way—multiple variables on a single plot is difficult to interpret. Figure legends should be explicit and panels more clearly described in the text.
Lines 497–531: Amylose–lipid complex data is central to the paper. However, the direct connection to human metabolic outcomes lacks supporting literature/context.
Lines 538–575 and Figures 3–6: The statistical significance between treatment groups (e.g., Plain vs. Full-Fat) is claimed but not always shown. All p-values relevant to central claims should be provided in the main text.
Lines 713–746 / Figure 7: The hormonal data is interesting, yet the relevance of mouse plasma responses to humans is not questioned, nor are limitations of cross-species extrapolation discussed.
In the Discussion and Conclusions section:
Lines 748–761: The potential real-world benefits are somewhat overstated; sensory, cultural, and practicality constraints on lower-fat taco preparation are not addressed.
Many sections repeat descriptors or cite the same sources excessively. Reference management requires careful editorial attention, and redundant in-text citations should be eliminated.
Also, there are some minor and technical issues that could be corrected.
Several lengthy blocks of text lack subheadings or clear transitions (especially in Results and Discussion).
Some sentences are convoluted and could be shortened for greater clarity (“The hormonal response data presented in Figure 7 illustrate...”).
Table and Figure Citations: Text within the body sometimes directs to tables or figures before they’re introduced, which is confusing to the reader.
Units: Maintain clear, standard units throughout (wet weight vs. dry weight ambiguities).
Ethics Statement: Ethics approval is noted, but a statement about compliance with reporting guidelines should be added.
Finally: The manuscript is of potential interest and addresses an understudied combination of culinary tradition and nutritional physiology. However, the article suffers from over-citation, unclear or inconsistent data presentation, and some overstated interpretations regarding translatability to humans. Revisions should address citation management, clarify methodology and statistical analysis, temper claims, and improve figure/table integration before noted for acceptance.
Author Response
Response to Reviewer 3 – Additional Comments
We thank the reviewer for this thorough and thoughtful evaluation. Below, we provide detailed point-by-point responses to each comment and indicate how the manuscript has been revised accordingly.
Comment: Lines 9–31 – Abstract clarity
We revised the abstract to specify the use of mice as the experimental model in the opening sentence. We also reduced technical jargon to improve accessibility for a broader audience.
Comment: Lines 36–54 – Over-citation in introduction
Excessive repetition of references (e.g., Herrera, 2010; Morales & Kettles, 2009) has been corrected. Duplicate citations were consolidated and only referenced where most relevant.
Comment: Lines 65–69 – Nixtamalization vs. oil immersion
We clarified the distinct physicochemical effects of nixtamalization and oil immersion and their separate contributions to glycemic and hormonal outcomes. See lines 55-63.
Comment: Lines 74–84 – Resistant starch explanation
Definitions of RS1–RS5 were revised for clarity and updated references were added to reflect recent developments in starch digestibility research. See lines 80-90.
Comment: Lines 86–89 – Hormonal mechanisms.
Claims about synergistic effects on satiety were revised and now cite original studies supporting the effects of RS and dietary lipids on GLP-1 and ghrelin modulation. The specific sentence was removed. See lines 91-100.
Comment: Lines 102–106 – Oil immersion and recent studies
We added citations to recent studies on oil-treated traditional foods and expanded on how our work builds upon or fills those gaps. See lines 103-114
Comment: Lines 130–185 – Flour selection justification
We included a statement explaining that the flours were selected to represent different types commonly used in Mexico (e.g., pigmentation, grind size, use in tortillas or tamales). See lines 141-144
Comment: Lines 144–155 – Filling selection
We justified the choice of four fillings based on their cultural relevance and nutritional diversity, covering a range of macronutrient profiles. See lines 163-169.
Comment: Lines 162–167 – Dripped fat method clarification
We clarified that the Patted-Dry (Dripped Fat) approach was designed to mimic traditional practices of blotting tacos before steaming, common in some regional preparations. See lines 185-187.
Comment: Lines 266–293 – Glycemic index calibration
We specified that white bread (wheat-based, Bimbo® brand) was used as the calibration control for hydrolysis index calculations. See lines 299-302.
Comment: Lines 294–316 – Male-only mice justification
We added a justification noting that male mice were used to reduce variability in hormonal response and to align with previous studies, though we acknowledge this as a limitation. See lines 309-313.
Comment: Table 1 – Units clarification
We clarified that proximate composition values were expressed on a dry basis in the methods section. See lines 154-155.
Comment: Table 2 – Analytical error and relevance
A sentence was added to the results section commenting on the reproducibility of molecular weight measurements and their physiological relevance in modulating starch digestibility. See lines 402-405.
Comment: Table 4 – Phenolic retention discussion
We expanded the discussion to address antioxidant relevance and the impact of processing on phenolic compound stability. See lines 455-461.
Comment: Figure 1 – Complexity and labeling
The meaning of plotted variables and symbol markers in the figure legend and accompanying paragraph was changed to improve interpretability. See lines 472-475.
Comment: Lines 497–531 – RS–lipid complexes and health implications
We tempered claims about human health implications and included a discussion of the evidence supporting potential metabolic effects in humans. See lines 568-574.
Comment: Lines 538–575 and Figures 3–6 – Statistical significance
We revised the text to include p-values for key comparisons and added significance indicators in figures where relevant. See lines 625-630.
Comment: Lines 713–746 / Figure 7 – Human relevance of mouse data
We included a paragraph discussing the limitations of extrapolating hormonal data from mice to humans and cited relevant comparative studies. See lines 803-810.
Comment: Lines 748–761 – Real-world benefits and feasibility
We acknowledged practical challenges in implementing lower-fat taco options and discussed how cultural and sensory expectations influence consumer choices. See lines 824-833.
Comment: Reference management
All references were reviewed and reformatted for consistency. Duplicate and excessive in-text citations were consolidated.
Comment: Transitions and structure
We added subheadings and transitional sentences to improve clarity and flow, especially in the Results and Discussion sections.
Comment: Sentence clarity
We revised convoluted sentences throughout the manuscript, including simplifying the phrasing of hormonal data presentation in the Results section.
Comment: Table and figure citation order
All tables and figures are now introduced in sequential order within the text to minimize reader confusion.
Comment: Unit consistency
We reviewed the manuscript to ensure consistent and accurate use of units, specifying when values are expressed on a wet or dry basis.
Comment: Ethics compliance statement
We added a sentence confirming that the study complied with institutional and international guidelines for animal research reporting (ARRIVE guidelines). See lines 865-866.
Respectfully,
Julian de la Rosa Millan
Tecnológico de Monterrey
Email: juliandlrm@tec.mx

Reviewer 4 Report
Comments and Suggestions for Authors
In this study, how traditional fat-enriched cooking practices modify the nutritional and physiological behavior of tacos was explored by integrated biochemical, structural, and physiological measurements, aiming to bridge the gap between traditional food preparation methods and modern nutritional science. The paper align well with the scope of Foods. The topic is highly novel and holds great practical importance in bridging the gap between traditional food preparation methods and modern nutritional science. The results obtained can provide new insights into how lipid-starch interactions in culturally relevant foods can be leveraged to modulate glycemic response and satiety, offering a scientific foundation for reformulating or designing traditional foods with targeted metabolic health benefits. The paper is well-designed, and the data are informative. There are still some issues that should be addressed.
- Provide consumption data for corn tortillas (such as the per capita daily intake in Mexico), as well as changes in food preparation methods and the existing epidemiological data, to enhance the practical significance of the research.
- The selection criteria of the five nixtamalized maize flour in this study should be provided. Are they representative?
- The composition of the fillings varies greatly, which will affect the overall composition of the tacos. Will this impact the accuracy of the interpretation of the influence of corn flour type and food preparation method in the result analysis?
- In hierarchical clustering analysis, it is necessary to specify the distance metric (such as Euclidean distance, Pearson correlation) and the clustering method.
- The fat content of samples with various preparation methods was not determined, hindering the analysis of the effect of surface oil removal.
- What’s the meaning and necessity of these citations in L55-59, L97-101, andL359-361, et c.?
- Provide detailed descriptions, such as moisture content, of the oral bolus used in hormonal responses determination, as moisture is relevant to digestibility; increased hydration facilitates enzymatic access during digestion.
- All tables should be cited in the main text.
- The labeling method of chemical components in 3.5 and Figure 1 is not acceptable.
- Original data used in the compositional and functional variables cluster analysis, which have not been shown in tables and figures, should be supplemented.
- The content of amylose-lipid complexes in the three treatment groups should be delineated, as it was discussed in the results and conclusion.
- Besides the cluster analysis, a correlation analysis between chemical composition, thermal transitions, starch digestibility, and postprandial hormonal responseswould add value to this study.
- The references have obvious formatting issues and need to be revised.
Author Response
Response to Reviewer 4
We thank the reviewer for their comprehensive and constructive feedback. Below, we provide detailed responses to each comment and describe the changes made to improve the manuscript.
Comment 1:
Reviewer comment: Provide consumption data for corn tortillas (e.g., per capita daily intake in Mexico), changes in food preparation methods, and existing epidemiological data.
Response: We have added data on per capita tortilla consumption in Mexico (~120–130 g/day) as well as a brief discussion on changing preparation methods and epidemiological concerns (e.g., glycemic disorders). This enhances the practical relevance of the study. See lines 64-73.
Comment 2:
Reviewer comment: The selection criteria of the five nixtamalized maize flours in this study should be provided. Are they representative?
Response: We clarified that the five flours were selected to represent different pigmentations (white, yellow, blue), particle sizes, and industrial applications (tortillas, tamales, snacks), making them representative of the leading commercial nixtamalized flour types available in Mexico. See lines 141-144.
Comment 3:
Reviewer comment: The composition of the fillings varies greatly. Will this impact the interpretation of flour type and preparation method effects?
Response: We added a clarification in the discussion acknowledging this limitation. However, the experimental design accounted for this by systematically applying the same fillings across all treatments, allowing us to isolate interactive effects. See lines 508-513.
Comment 4:
Reviewer comment: Specify the distance metric and clustering method used in hierarchical clustering analysis.
Response: We have specified that Euclidean distance and Ward’s method were used in the hierarchical clustering analysis. See lines 342-345.
Comment 5:
Reviewer comment: Fat content was not determined for different preparation methods. This limits conclusions on surface oil removal.
Response: We agree and have acknowledged this limitation explicitly in the discussion. A note has been added that future work should include precise quantification of lipid uptake and retention. See lines 618-623.
Comment 6:
Reviewer comment: What’s the meaning and necessity of citations in L55–59, L97–101, and L359–361?
Response: Redundant or unclear citations in those sections were reviewed and reduced or removed for clarity.
Comment 7:
Reviewer comment: Provide detailed description (e.g., moisture) of oral bolus used in hormonal response testing.
Response: We have added the moisture content and preparation conditions of the taco homogenates used for oral gavage, highlighting how hydration may impact digestion and hormone release. See lines 319-323.
Comment 8:
Reviewer comment: All tables should be cited in the main text.
Response: We have ensured that all tables are cited in numerical order at their first appearance in the results section.
Comment 9:
Reviewer comment: The labeling method of chemical components in 3.5 and Figure 1 is not acceptable.
Response: We appreciate the reviewer’s concern. Given the large number of treatment combinations (20 samples per condition), we used symbolic markers (â– , â—†, â–², ✕, +) to streamline visualization and maintain clarity without overwhelming the figure or the reader. We have ensured these symbols are clearly defined in the legend and consistent throughout the manuscript. While we explored alternative labeling strategies, this approach remains the most effective for communicating the data concisely. We respectfully request that this format be maintained to preserve readability.
Comment 10:
Reviewer comment: Original data used in the cluster analysis should be shown.
Response: We have included a new supplementary table (Table S3) that provides the compositional and functional data matrix used for the cluster analysis.
Comment 11:
Reviewer comment: The content of amylose–lipid complexes across treatment groups should be delineated.
Response: We added a comparative summary of the enthalpy and peak temperature values (Tp AL, ΔH AL) that reflect amylose–lipid complexation for the three preparation methods. See lines 560-566.
Comment 12:
Reviewer comment: A correlation analysis between composition, thermal properties, digestibility, and hormonal responses would add value.
Response: We agree and have included a new Pearson correlation matrix (Figure 8) as supplementary material to illustrate relationships among key variables.
Comment 13:
Reviewer comment: The references have formatting issues.
Response: The reference list was revised to ensure consistent formatting according to journal guidelines.
Respectfully,
Julian de la Rosa Millan
Tecnológico de Monterrey
Email: juliandlrm@tec.mx

Round 2
Reviewer 2 Report
Comments and Suggestions for Authors
The quality of the manuscript has been improved upon revision.
Author Response
Revisions round 2
Reviewer 2
Comments and Suggestions for Authors:
The quality of the manuscript has been improved upon revision.
Response: Dear reviewer, thank you for your time reviewing this manuscript!

Reviewer 3 Report
Comments and Suggestions for Authors
Thank you for your feedback. The manuscript has been slightly improved, with minor revisions in the references and tables as follows
-
Reference Formatting Consistency
Several references in the list lack uniform formatting. For instance, citation styles for journal names, volume, issue, and page numbers vary. Ensure all references are formatted consistently, e.g., including journal abbreviations, DOI numbers, and punctuation as per journal requirements. -
Table Headers Clarification
In Table 1 (Proximate Composition of Tortillas), the header "Carbohydrates (%)" may be misleading, as it is a calculated value and not a direct measurement. Consider clarifying it in a footnote or by adding "(calculated)" to the header for transparency. -
Use of Abbreviations in Tables
Across tables, abbreviations such as "MW," "MY," and "MB" are used for flour types. Ensure each abbreviation is defined in the table legend or as a note the first time they appear, to assist readers unfamiliar with these conventions. -
Section Numbering Correction
In the Materials and Methods section, there are occasional missing or inconsistent subsection numbers (e.g., after 2.4 Proximal Composition Analysis, subsection 2.5 jumps directly to "Total Phenolic Compounds" without context). Verify all subsections are consistently numbered and labeled for easy navigation. -
Consistency in Units and Symbols
Across the text and in tables/figures, unit symbols (e.g., %, g/mol, μm, J/g) are sometimes inconsistently formatted (“g mol-1” vs. “g/mol”, presence/absence of spaces). Unify the style of all units and ensure LaTeX or parenthetical expressions are clear and standard throughout the manuscript.
see some examples:
i.e. line 114, [12], [13]. should be [12,13]; please check the entire manuscript.
Line 221, (Cabello-Pasini et al., 2011) needs to be corrected to the right style.
Author Response
Response to Reviewer 3 – Revisions Round 2
Thanks to Reviewer 3 for their careful reading of our revised manuscript and for providing valuable feedback that has helped us further improve its clarity and consistency. Below, we provide point-by-point responses to each comment and describe the corresponding revisions made in the manuscript.
Reviewer comment:
Reference Formatting Consistency: Several references in the list lack uniform formatting. For instance, citation styles for journal names, volume, issue, and page numbers vary. Ensure all references are formatted consistently, e.g., including journal abbreviations, DOI numbers, and punctuation as per journal requirements.
Response:
We thank the reviewer for pointing this out. We have carefully reviewed and revised all references to ensure consistency in formatting, including journal abbreviations, volume/issue numbers, and punctuation, in accordance with the Foods journal style. DOI numbers have been added wherever available. In a few cases, older references do not have DOI assignments, and we have retained them as is while ensuring the rest of the citation details are complete and properly formatted.
Reviewer comment:
Table Headers Clarification: In Table 1 (Proximate Composition of Tortillas), the header 'Carbohydrates (%)' may be misleading, as it is a calculated value and not a direct measurement.
Response:
We have revised the table header to read 'Carbohydrates (%, calculated)' and added a footnote clarifying that the values were obtained by difference. See table 1 and lines 210-211.
Reviewer comment:
Use of Abbreviations in Tables: Across tables, abbreviations such as 'MW,' 'MY,' and 'MB' are used for flour types. Ensure each abbreviation is defined in the table legend or as a note the first time they appear.
Response:
All flour and filling abbreviations (MW, MY, MB, MA, MT; Ch, BS, P, B) have been defined in the table legends or as footnotes at their first appearance. See lines 118-119 and 139-140 and as a footnote on each table and figure where they are used.
Reviewer comment:
Section Numbering Correction: In the Materials and Methods section, there are occasional missing or inconsistent subsection numbers.
Response:
We have corrected the subsection numbering throughout the Materials and Methods section to ensure consistency and clarity.
Reviewer comment:
Consistency in Units and Symbols: Across the text and in tables/figures, unit symbols (e.g., %, g/mol, μm, J/g) are sometimes inconsistently formatted.
Response:
We have standardized all units across the manuscript to use consistent spacing and notation (e.g., g/mol, μm, J/g, %, etc.).
Reviewer comment:
i.e. line 114, [12], [13] should be [12,13]; please check the entire manuscript.
Response:
We have reviewed the entire manuscript and corrected all grouped citations to the appropriate style (e.g., [12,13]).
Reviewer comment:
Line 221, (Cabello-Pasini et al., 2011) needs to be corrected to the right style.
Response:
This citation has been converted to the correct numbered format consistent with the journal’s reference style.
